# Underlying Music Mechanisms Influencing the Neurology of Pain: An Integrative Model

**DOI:** 10.3390/brainsci12101317

**Published:** 2022-09-29

**Authors:** Joanne Loewy

**Affiliations:** The Louis Armstrong Center for Music and Medicine, Mount Sinai Health System, Icahn School of Medicine, New York, NY 10003, USA; joanne.loewy@mountsinai.org

**Keywords:** music therapy, pain management, chronic pain, palliative music, music medicine

## Abstract

Pain is often debilitating, and is associated with many pathologies, as either a cause or consequence. Pharmacological interventions, such as opioids, to manage pain may lead to potential problems, such as addiction. When pain is controlled and managed, it can prevent negative associated outcomes affiliated with disease. Music is a low-cost option that shows promise in the management of painful circumstances. Music therapy has provided potent options for pain relief across a variety of ages and populations. As a nonpharmacological alternative or complement lacking side effects, music interventions are growing in clinical application and research protocols. This article considers the neurological implications of varying kinds of pain to provide working considerations that preempt the use of music and music-therapy applications in treating pain.

## 1. Introduction

Pain can be a debilitating condition and is often associated with disease progression. Music, which is comprised of organized sounds and silences, has been influential in the treatment of all kinds of pain since virtually the beginning of recorded time. While various kinds of pain can be helpful in diagnosing illness, chronic pain is a common complaint of patients of all ages and diagnoses, affects more than one in four individuals, and is associated with numerous pathologies, as either a cause or a consequence [1]. Music therapists are involved in the clinical applications of sound and music, and as such are poised to understand the discrete mechanisms involved in both the efficacy and feasibility of music interventions. Music therapists are trained to assess and evaluate the implications and outcomes of music applications, whether applied as recorded or live, in real-time applications. This is significant, as increasingly research indicates that the implementation of prescribed music within an incident of pain, whether acute, chronic, or procedural, can be, particularly for those experiencing pain, contraindicated and actually do harm [2].

This article considers the neurological implications involved with varying kinds of pain and provides working considerations that preempt music and music-therapy applications in treating pain. Critical to the understanding of neural influences in pain perception will be consideration of both physical and emotional substrates informing the evaluation and assessment of pain.

## 2. Assessment-Integrative Mechanisms

The inclusion of critical elements necessarily formulates a gestalt of how a pain experience is understood. In such classification, a clinical interview is mandatory. The Pain Assessment Quadrant (PAQ, see below, Figure 1) reflects essential elements that can be included within the assessment. The patient’s voice and expression of pain is most advantageously and accurately experienced live. The PAQ involves the therapist’s use of self, inclusive of the therapist’s transference and/or countertransference.

A multidisciplinary or integrative consultation involving the patient’s team of medical and supportive professionals and involved family members provides a psychosocial historical context of the patient’s background. Finally, an ever-broadening understanding of the literature and findings necessarily and specifically relate to pain treatment outcomes, and these are disease-specific to the patient’s diagnosis and their experience of pain. Each of these aspects, seen as separate and when considered in combination as well, inform the adherence to best-practice options for the patient experiencing pain.

### 2.1. Definition of Pain

In 1979, the Subcommittee on Taxonomy, adopted by the IASP (International Association for the Study of Pain) council cited a formal definition of pain as the following: “An unpleasant sensory and emotional experience associated with actual or potential tissue damage, or described in terms of such damage”.

In 2018, the same organization updated its definition of pain, defining it as “An unpleasant sensory and emotional experience associated with, or resembling that associated with, actual or potential tissue damage” [3], In an unusually descriptive diagram (Figure 2) outlining their subcommittee’s finding of a qualitative investigation, the association placed greater emphasis on the patient experience, adding the text: “resembling that associated with” to their definition. This implies that pain’s effects of any magnitude, whether or not practitioners know the exact etiology, should be believed. Pain has a sensory and emotional component. The IASP has been influential in how practitioners understand pain.

### 2.2. Theories of Pain

#### 2.2.1. The Gate Control Theory

The gate control theory of pain’s important contribution to the understanding of pain was its emphasis on central neural mechanisms. The theory prompted medical and biological sciences to accept the brain as an active system that filters, selects, and modulates inputs. The dorsal horns, too, were then accepted as not merely passive transmission stations but actual sites whereby dynamic activities (inhibition, excitation, and modulation) occur [4].

The challenge of this theory was in understanding how the brain functions, and specifically, how the “gates” could be monitored. The neuromatrix theory furthered understanding of such processes. As the brain possesses a complex system of neural networks, the relationship between the body and self constructs a neuromatrix that integrates multiple inputs to produce an output pattern that evokes pain.

Following an injury, pain signals are transmitted to the spinal cord and then up to the brain. Melzack and Wall suggest that before the information is transmitted to the brain, the pain messages encounter “nerve gates” that control whether these signals are allowed to pass through to the brain [4].

In some cases, the signals are passed along more readily than others, and pain is experienced more intensely. In other instances, pain messages are minimized or even prevented from reaching the brain at all. This gating mechanism takes place in the dorsal horn of the body’s spinal cord. Both small nerve fibers (pain fibers) and large nerve fibers (normal fibers for touch, pressure, and other skin senses) carry information to two areas of the dorsal horn [4]. These two areas are either the transmission cells that carry information to the spinal cord and to the brain or the inhibitory interneurons, which halt or impede the transmission of sensory information.

#### 2.2.2. Neuromatrix

Ronald Melzack updated how pain is perceived by including a “neurosignature” output of the neuromatrix. Included patterns of nerve impulses of varying temporal and spatial dimensions are thought to be produced by neural programs genetically built into the neuromatrix. These determine the particular qualities that inform the pain experience and other properties associated with behaviors related to pain [5].

The body–self neuromatrix comprises a widely distributed neural network that includes parallel somatosensory, limbic and thalamocortical components that subserve the sensory-discriminative, affective-motivational, and evaluative-cognitive dimensions of pain experience. The synaptic architecture of the neuromatrix is determined by genetic and sensory influences.

Multiple inputs that act on the neuromatrix programs and contribute to the output neurosignature include sensory inputs (cutaneous, visceral, and other somatic receptors), visual and other sensory inputs that influence the cognitive interpretation of the situation, phasic and tonic cognitive and emotional inputs from other areas of the brain, intrinsic neural inhibitory modulation inherent in all brain function, and the activity of the body’s stress-regulation systems, including cytokines and endocrine, autonomic, immune, and opioid systems [5].

These multiple systems have a theoretical framework that is distinctly determined by a genetic template designated by a body–self relationship manipulated by how we regulate our stress system and additionally, our cognitive brain function, which is inclusive of sensory inputs.

### 2.3. Types of Pain

Pain can be experienced as sharp, piercing, dull, or achy. Sharp pain is often experienced as sudden and acute, while chronic pain can be tonic and recurrent. In general, pain is characterized as nociceptive, inflammatory, neuropathic, or functional. While nociceptive pain is typically the result of tissue injury, inflammatory pain is associated with inflammation caused by an atypical response by the body’s immune system. Neuropathic pain’s etiology involves nerve irritation, while functional pain is routed from an unknown origin. It is unexplained pain that comes from a source.

A pain experience is often linked to anxiety [6]. Experimental studies of animal models have considered music’s potential in pain reduction, though the discrete effects are undefined and have not added much evidence toward supporting the development of defined mechanism-based music interventions in human clinical settings [7].

Interestingly, music has been shown to influence glutamate secretion and glutamatergic neuronal response, and can modulate activity in limbic and paralimbic brain structures, including the ACC (anterior cingulate cortex). The ACC is susceptible to music’s influence, as it is uniquely positioned in the brain and connects to the limbic system and the prefrontal cortex [8]. As such, instituting music therapy to address emotional response, which is part of the pain experience, poses a plausible, risk-free intervention that can influence neural correlates associated with emotion.

### 2.4. Music Therapy’s Access to Neural Mechanisms with Clinical Features across Developmental Stages

A customary way that researchers of music medicine and music therapy distinguish the implementation of music, is to characterize it as “active” and “passive”, and/or “active” and “receptive”. This idea, prompted by a review [9] and others as well, defines active music therapy as the involvement of a music therapist, implying interactive communication. Passive music therapy, on the other hand, involves—under these terms—listening to music for a particular purpose, recorded or live, without the involvement of a music therapist. Listening to prerecorded music, however, has also been purported as music medicine in contrast to active music therapy [10] and yet music therapists do implement recorded music listening experiences into their work at times as well.

The problem for such definitive yet misinterpreted distinctions becomes more complicated when the involvement of music mechanistic thinking comes into play. For instance, one can be listening to music and become quite activated in the neurological sense. At the same time, in playing music, it is possible to be quite passive and uninvolved, or even “tuned out” during what might appear to resemble apparent expression or activation.

A recent study [11] showed that when compared to listening to music with a fixed tempo, an entrained response was key in effective neural modulation. Entrainment when compared to a fixed tempo rendered a more effective strategy toward achieving a psychophysiological relaxation response.

This occurs within a dynamic acutely synchronized process of live music paired with an internal body rhythm, which increases peripheral blood flow. Interestingly, the entrained music experiential utilized in this particular study was the humming of a pentatonic scale, performed by a music therapist. The musical beat implemented was initially and strategically synchronized exactly to the participant’s pulse [11]. After three minutes, the music was slightly and gradually decelerated to evoke a relaxation response via entrainment.

The body’s response to vibration has been categorized into hemodynamic, neurological, and musculoskeletal. Hemodynamic effects include stimulation of endothelial cells and vibropercussion; neurological effects include protein kinase activation, nerve stimulation, specifically vibratory analgesia, and oscillatory coherence; musculoskeletal effects include muscle stretch reflex, bone cell progenitor fate, vibration effects on bone ossification and resorption, and anabolic effects on spine and intervertebral disks [12].

I have used vibration of dissonant to consonant intervals in my work with pediatric and adult patients and have seen profound results, most particularly with the gong, while toning, and in instituting breath entrainment incorporating a technique I developed called tonal intervallic synthesis, which implements (Suzuki) tone bars to assess consonant and dissonant pain response [13]. This involves the movement of held vibratory tones played and scanned over the entire body to start, and then the intentional application of specific intervals on areas of pain—moving from dissonant sung and played tones—assessing resultant audible overtones, to consonant intervals and audible overtones. One major question in neuroscientific music research is whether musical pieces evoke specific subjective emotional and arousal reactions, which are related to neurophysiological activation patterns [14]. As music can affect thinking and feeling and is accessible through conscious and unconscious vibratory, auditory, and emotional systems, it is useful to address how specific music mechanisms can influence neurological processes distinctly, and explicitly throughout the life span.

#### 2.4.1. Neonatal Pain

In the womb, the neonate begins hearing as early as 19 weeks [15]. The vibratory patterning of a mother’s heartbeat is computed and the infant’s developing brain likely entrains to the rhythm, prebirth.

The infant’s impressionable neural plasticity and the impact that silences and sounds may take on physiologically in brain development determine vitality [16]. Music therapeutic applications are making strides in validating how models of neurobehavioral care influence self-regulatory behaviors.

When neonatal neural structures are compromised or lacking in auditory order, plasticity can be threatened, resulting in cerebral cortex and auditory system impediments. Further development in subsequent years may need to catch up or further adapt to accommodate sensory neural processing and when responding to environmental changes [17].

We have witnessed the strong propensity of rhythm through study of neonates in a variety of states, including pain. Furthermore, the influence of culturally familiar melodic sequencing has enhanced opportunities for attachment and bonding, particularly during painful procedures [18]. The use of familial melodies and/or favorite themes of parents, either passed down from forefathers and mothers or created within a music-psychotherapy session context, can trigger meaningful emotive physiological responses in parents that are experience by neonates. “Song of kin” can calm parents and infants and may help stabilize an infant through painful procedures [19]. Parents’ voices and melodies that render emotional familiarity as sung in the moment with vocal intention and that reference a current or former pleasant memory can translate as safety for the neonate [19] and can stabilize the risk of a stress response in parents [20].

The first week of life for neonates postbirth is a notable time of stress for neonates. The NISS (Neonatal Infant Stressor Scale) is used to rate neonatal fragility and stress characteristics, for example, that are associated with procedures. Ratings reflect that underlying associations and adverse neurodevelopmental outcomes can result when preterm infants’ somatosensory circuitry in the spinal cord is threatened due to early life experiences [20] Researchers such as van Dokkum and her colleagues have shown how repeated handling, painful procedures, and all other stressors combined likely contribute to overstimulation of the central nervous system [21].

In studying the mechanisms of music characteristics [16], culture necessarily will determine the music that is best indicated for clinical interventions. Whether a meterless Indian raga, or a baroque piece, or a lullaby perhaps phrased in 3/4 or 6/8 time, understanding the music of a parent’s background is of seminal importance, particularly with neonates [18].

Access to the tunes infants heard in utero and presenting this music live and entrained, in a format that parents can utilize as their “song of kin” [18] may ensure continuity of care (in the NICU, and at home, postdischarge). Such reliability likely will influence the potential for bonding as an incentive for what some researchers have studied to be a “neurologic brain effect” [16] in premature infants.

#### 2.4.2. Pediatric Pain

Pediatric pain has been addressed in multiple contexts, but perhaps most in the occurrence of procedural pain. This is likely because children, moving from sensorimotor to concrete operational stages of development can readily begin to participate in the musical tools provided to them by music therapists to mitigate pain. One study showed how children’s concrete notions of pain were more readily accessed in the preteen years [22].

Understanding amongst health professionals and parents as to how children of varying ages and backgrounds, along with study of the unique characteristics that accompany and typify developmental stages, provide critical information [23]. Learning from children themselves through play [24,25] and observing how they express and cope with pain experiences using creative options, such as music [25,26], should be mandatory, as such behaviors help clinicians ascertain the most culturally sensitive music that will ultimately help the medical and psychosocial teams in effectively assessing and managing all kinds of pain in children.

Structured ensemble play where children are purposefully invited to express themselves melodically and rhythmically is experienced in the brain. Such playful expression when framed within harmonic support from a music therapist creates a trusting relationship, and such an achievement can often forgo the impact of the pain response.

In cases involving the precise moments of pain crises, music expression fosters connection within a relationship that may have important implications for children’s endurance of other kinds of pain in later hospitalizations, particularly when experiencing an acute pain crisis. A recent study analyzing the role of trust in the care of young children who were diagnosed with diabetes [27] emphasized what health-care providers have found to be most essential in trust-building, particularly with regard to issues associated with pain. The study emphasized the impression that pain made on young children and how “starting off right” is important in reducing the risk of excessive fear and needle anxiety.

The foundation for a building of resilience begins with trust and explicit honest procedure explanation, rather than manipulation, such as distraction [28]. Drumming can encourage release, and when timed with a trusted therapist can alleviate stored tension of pent-up energy that exacerbates the pain experience [25]. Drumming has been utilized in painful ancient rite-of-passage experiences, such as fire walking and body piercing.

#### 2.4.3. Adults

It is not a secret that numerical pain ratings often fail to convey the complex nature of the pain experience [29]. Whereas research on pain in children shows evidence that pain is underrecognized and undertreated [30], adults’ experience of pain can be overreported or under-reported with greater cognitive sophistication in their reasoning of potential outcomes.

In a recent exploratory study of 338 patients, researchers examined self-reported motivations for why people over- and/or underreport their own pain [31]. Shifts in their pain responses were based on their perceptions involving internal factors related to negative affect, and contextual factors such as a lack of interpersonal trust and expectations of perceived biases of physician perceptions and treatment. An aversion to stigmas associated with pain was also a concern.

Motivations for modulating their own pain reports relative to a numerical pain scale (0–10) highlighted in this research revealed overreporting pain to gain more immediate relief options from staff. Concerns related to underreporting included financial worries and a perception of how a patient might be perceived by others. Issues of trust in providers and personal ethics were mentioned frequently [31]. Recognition of patient concerns and one’s own personal biases toward others’ pain reporting is an outcome of the numerical rating scale study that recognized that illuminating such factors may improve patient–provider trust and additionally more effectively support precision of the usage and interpretation of numerical pain ratings.

The largest review of music’s effect on procedural pain recovery for adults was undertaken in 2015 [32], and in its evaluation of mechanisms, it acknowledged that pain experience is affected by both physical and psychological influences. The review indicated that listening to music can alter the actual intensity and unpleasantness of pain, which in turn can reduce the sensation of pain to be reduced. Additional potential mechanisms could result in reduced autonomic nervous system activity, such as reduced pulse, respiration rate, or decreased blood pressure.

Areas of study in music therapy for pain in adults occur most frequently in palliative care and in debridement [33,34,35]. Surprisingly, it is common for studies to leave out patient culture in the assessment of how to treat pain. Cultural music preferences, as opposed to song lists or even “genre preference”, is seminal to the practice of patient-centered care. While music is central, studies that leave out the inclusion of a music therapist whose training involves entrainment, and the understanding of the symbolic association of instruments [36,37], psychosocial constructs and clinical improvisation [37], might be missing critical opportunities. Few references discuss gonging and vibration, which we have found to be seminal in helping patients control their blood flow and mood state [26,27].

In investigating any kind of pain, whether it be acute, chronic, or procedural, it is recommended to never overlook the relationship between pain and anxiety. They are interchangeable and often indicative of strong clinical prowess when they are addressed together. In doing so, elevated stress levels, depression, insomnia, and resultant acute and/or posttraumatic stress in treatment planning can be avoided [6].

A recently published feasibility trial implements a music-assisted relaxation (MAR) protocol using entrained live music combined with guided relaxation and/or the use of imagery [35]. The effects of the MAR were compared to a control group (treatment as usual) over a period of 2 weeks maximum or six interventions. This protocol investigates the effects of a live MAR protocol provided by a certified music therapist on background pain, anxiety, and depression, vital signs and medication intake in adult burn patients. MAR can elicit a relaxation response, which when paired with creative thinking and visualization, might enhance resource options for pain management as part of a treatment regime.

A majority of studies on pain and music implement recorded music and focus on particular kinds of pain defined by etiology. Most often, the implemented recordings are defined by style and genre, rather than by elements, characteristics, or mechanisms [38]. In emphasizing the importance of music selection and that music application be scrutinized and analyzed for mechanistic elements, Martin-Saavedra et al. [39] consider the negative ramifications in the usual overgeneralization of “genre”, even as composer-based music is often indicative of researchers’ earnest aims to provide indicated music. Assumptions made about music often render limitations in subsequent outcomes and result in restricted applications for treatment.

One example highlighting such complexity in music clinical decision-making relating to the characteristic features of music application was undertaken by Gutsgell [40]. In implementing distinct music therapy mechanisms related to adult patients in palliative care who experienced pain, the institution of choice with silence or water-flow sounds (ocean drum) and centering with subtle directives indicated careful assessment. This, followed by the succession of distinct layering of sounds—melody, harmony, meter shifts, and key modes (mixolydian to start)—was notable and stood out as it instituted in-the-moment evaluation, rather than a prescriptive type of music named broadly as a genre, or composer.

#### 2.4.4. Chronic Pain

Pain chronification describes the process of transient pain progressing into persistent pain. It is interesting to consider some of the research that has delved into looking at factors that prolong or intensify the experience of pain, and how pain’s endurance and intensity can vary from one person to the next, even amongst patients of similar age, sex and diagnosis.

One study of 611 patients investigated persistent postmastectomy pain. Strangely, treatment-related factors including surgical type, surgery complications, axillary node dissection, recurrence, tumor size, radiation, and chemotherapy were not significantly associated as influencing the reported pain, postmastectomy. Rather, persistent pain was affiliated with psychosocial variables, such as catastrophizing, somatization, anxiety, sleep, and depression, were mentioned as considerably more determinant of how pain was experienced [41].

Bringing creativity into treatment as a central component and as part of a dynamic relationship, affords options for pain management, and most particularly can spark “out of the box” ways of addressing chronic pain. Creativity, for instance, a process involved in clinical music improvisation, can include ritual, but also has inferences of suggested unconventional pathways of play, which can loosen the effects of what otherwise would be translated as a traumatic experience [42]. Creative music therapy in pain experiences calls upon clinicians to implement ingenuity and resourceful ways of being [43].

Research that involves such processing includes the coupling of disparate brain regions. Creative thinking and resourcing can be an essential aspect of working through discomfort. Specifically, creativity often involves coordination between the cognitive control network, which is involved in executive functions such as planning and problem-solving, and the default mode network, which is most active during mind-wandering or daydreaming [44] The cooperation of such networks may be a unique feature of creativity. These two systems are usually antagonistic. They rarely work together, but creativity seems to be one instance where they do [45].

Some research suggests that spending time in nature can enhance creativity. The natural world’s capacity to restore attention, or in other circumstances its influence in letting the mind wander [46], makes the great outdoors an ideal place to spark creativity.

#### 2.4.5. Palliative Care

In treating pain in palliative care, it is not uncommon to come across studies that do not include patient-selected music. Equally as disappointing is the lack of music therapists as the interventionists or even consults, in palliative care [47]. Such omissions likely have missed a potential for the outcome of what could have influenced a music effect in pain management.

A recent study assessed the prevalence of sleep disturbances and possible correlations with associated factors in a large number of patients with advanced cancer who were admitted to different palliative care settings [48]. Of the 820 patients receiving palliative care, more than 60% were found to have relevant sleep disturbances. The authors concluded that physicians can make useful clinical inferences related to sleep disturbances and take more subsequent treatment measures to address pertinent factors associated with lack of sleep.

Sleep, as a desired outcome for enhancing comfort in treating pain, is a useful option. However, not much has been reported on its serving as a catalyst for enhancing creativity, especially in painful crises of disease processes. This may have important ramifications for music therapists who treat people whose pain is severe to the point where they have insomnia.

When experiencing painful episodes, or at times of transitioning from alive toward passage moments moving toward death, or even when working to palliate a particularly painful acute crisis, having a sedation protocol may serve as a potent resource for treatment. Loewy has developed a sedation protocol that was first studied on infants and toddlers [49] and has since been used with adults [50] in hospital care.

Sleep loss can have a negative effect on disease recovery, with neurological implications. One study showed a correlation between underlying brain and behavioral mechanisms explaining the common occurrence between poor sleep and pain perception [51]. In particular, sleep deprivation was shown to increase pain in the brain’s cortex sensory spots, but also showed shut-off sensitivity to the spots that modulate pain processing, namely, the striatum and insula. This study also showed the important role of regulation and predictability with sleep, as changes in duration and timing may affect how pain is processed.

When rest can come into practice, problem solving and negotiation on a concrete and cellular level can expand [52]. Protocols for inducing imaginative thinking and visualization may be preemptive as a sleep tool. Concentration and utilizing entrainment to alter the pain experience usually begins with breath extension and body relaxation techniques. This can be helpful, particularly when timed with procedures, such as venipunctures [53].

#### 2.4.6. Anxiety and Pain

Listening to patient-preferred favorite music preoperatively, while it does not necessarily reduce pain, can reduce anxiety, and this has been found in a study of 117 adult patients ages 17–70 years old that received elective inguinal hernia surgery. The music listeners findings correlated with regulation of hemodynamic parameters, and improved postoperative patient satisfaction. However, reduced anxiety was not associated with reduced pain [54].

This points to speculation that while people might experience pain related to procedure or illness, resilience can be built through music-which can lessen the anxiety response [53] that in therapy-supported contexts would usually likely result from pain. In a growing number of studies, anxiety can be reduced, even where pain might not be lessened.

### 2.5. MCS (Music Characteristic System)

Patient-selected music is thought to be most opportune in treating patients of all ages and diagnoses because it is recognized as familiar and therefore considered comforting, particularly in treating pain.

What is often left out in music implementation in research and practice are considerations of dosage, elements of activation or consideration of how downregulation can be predicted, and inquiries that inform the therapists administering the music, or what memories or associations are assimilated with a “preferred” tune or song.

The MCS ([55], Figure 3) can be used to inform music therapists of the unique mechanisms that will best lead toward the development of an “audio elixir” program, whether live or recorded. The music used in a pain session, for instance, when analyzed according to an MCS, can be characterized through identifying numerical values and elements that reflect the overall qualities of sequential changes in the music.

The numerical values are arranged from low to high, with lower values identifying the music as less “active”, while higher values indicate that the music is more “active”. Less active music may equate to what is generalized as being “relaxing” and more active music as being “activating”. Extra musical elements offer insight as to perceived or felt emotions as they come up and are experienced by the listener. In this way, images and metaphors are defined and considered.

The Music Characterization System (MCS) ([55], Figure 3) helps to identify a sequence in which to play music, which can inform how pain interventions that provide therapeutic value are instituted. Its guidelines seek to focus attention to accurately match prerecorded or live music to address desired clinical goals. Music, when selected by patients and then analyzed using 12 musical elements that then obtain a numeric rating of 1–10 in order of higher/decreasing activation and arranged in a sequence of “highest activation” to “lowest activation” can invite therapeutic predictability and thus provide a trusting relationship between the therapist and patient.

## 3. Conclusions

While pain was mandated over four decades ago by the Joint Commission on Accreditation of Healthcare Organizations to be distinguished and regulated as “the fifth vital sign” and one that would require caregivers to regularly assess and address [56], there is still limited literature on approaches in general and nonpharmacological treatments specifically addressing pain. For quite some time, neonatal and infant pain was denied in the medical community. Thanks to the persistent and compelling research of Kanwaljeet Anand [57], the American Academy of Pediatrics and the Canadian Paediatric Society released a statement to increase awareness that neonates do in fact experience pain. Their admission provided a physiological basis for pain assessment and management, and furthermore recommended reducing the exposure of neonates to noxious stimuli and treating neonatal pain with effective and safe interventions [58].

Behaviorism teaches that reduction as a goal on its own is not realistically obtainable without having something in place that is rewarding and of interest to a consumer. However, “reducing pain” is the wording of most studies and clinical practices that fall under the category of pain management in health care.

This article has defined some of the underlying music mechanisms involved in the pain response, particularly related to assessment, expression, and treatment. Themes of pertinent research have elucidated some of the seminal issues related to treating people of varying ages and diagnoses. Creativity and the fostering of unique experiences that can be elicited by both live and/or recorded music have been highlighted.

The institution of individualized patient evaluation, most particularly with dynamic music options, should begin with no assumptions, but rather listening to the patient’s voice and expression of content. There is no doubt that treating a person who is experiencing pain of any etiology requires careful, thorough, individualized assessment. It is my hope that clinical music medicine or music-therapy interventions and research meant to address a person’s pain experience will include assessment of the range or limitation of affect as well as an analysis inclusive of the patient’s rhythm of speech, whereby release or constriction may lead toward offering the most effective music options. Team influence and family inquiry may also help define the discrete, distinct needs of a person’s pain episode. No two pain episodes are exactly alike, which is precisely why music therapy holds unique properties that not only have been recognized through transcripts since the beginning of recorded history, but will likely continue to hold important outcomes for the practitioners and patients who will benefit from its inclusion as a complementary modality or as an alternative prior to pharmacological standards, or when other interventions have been exhausted.

## Figures and Tables

**Figure 1 brainsci-12-01317-f001:**
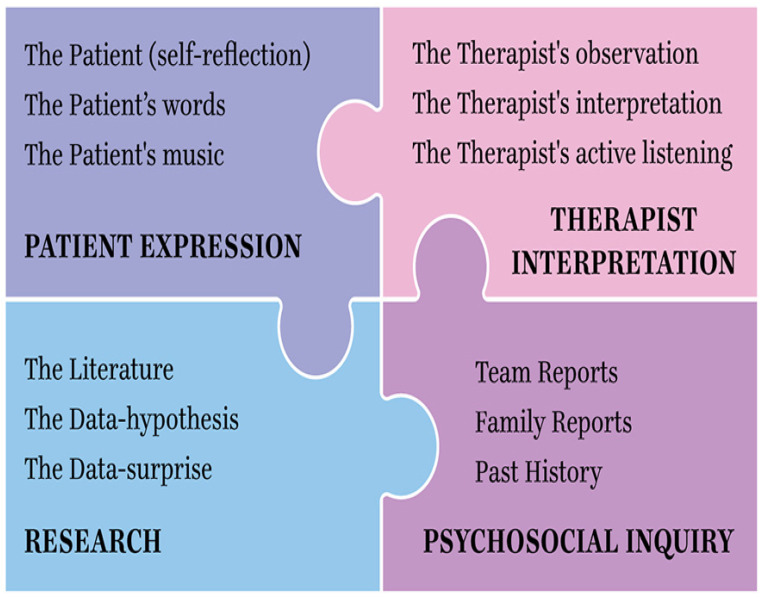
Pain Assessment Quadrant: an integrative model.

**Figure 2 brainsci-12-01317-f002:**
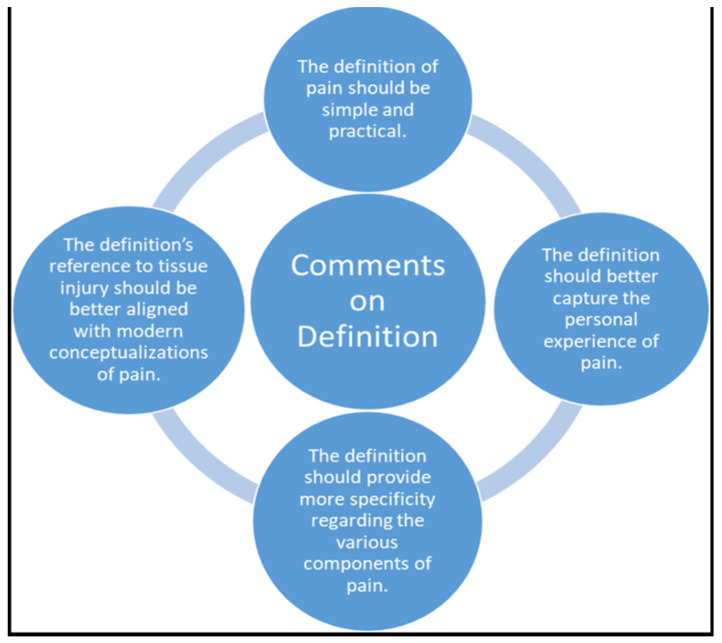
Comments on the Definition of Pain (IASP, 2018).

**Figure 3 brainsci-12-01317-f003:**
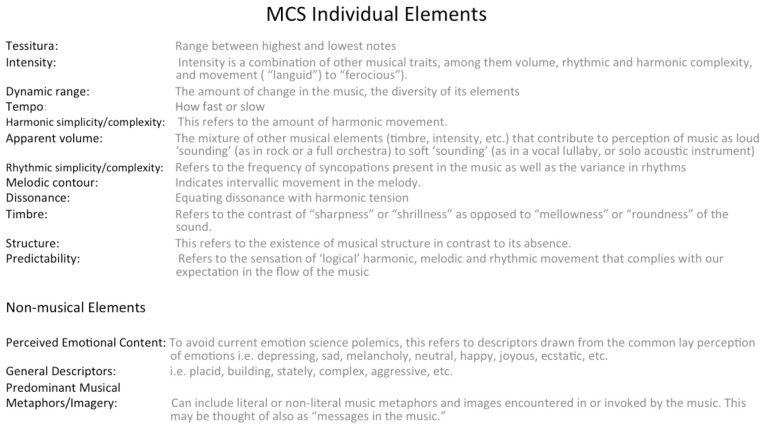
MCS [55].

## Data Availability

Not applicable.

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
