# Peer review of "Underlying Music Mechanisms Influencing the Neurology of Pain: An Integrative Model"

_brainsci, 2022, doi:10.3390/brainsci12101317_

Round 1
Reviewer 1 Report
This is a fabulous paper, meeting a need in the field, in terms of explaining current research and practice of music therapy as part of pain care. This paper will be terrifically useful for clinicians, healthcare managers and music therapists. I strongly recommend publishing. The list below are mainly typos and a few suggestions for structure and content that might improve the reader's experience of the paper.
2.2.1 Gate contro (typo need l)
p4 line 119 A pain experience is often linked to anxiety - reference this claim
p4 line 132 - 133 I wonder about just using the terms active and receptive rather than passive? Do you need both terms passive and receptive?
p 5 paragraph starting line 164. The use of 1st person here is jarring. Up until now the whole section is factual and descriptive. Suddenly the author says 'I have...' I wonder if this should come later, or be written in 3rd person still to match this whole section which is presenting evidence and prior information.
2.4.1 neonatal pain
While I can see how it makes sense to start with neonates and work through the age groups, this section feels a little out of place. There is rare mention of pain in this section, it seems more about working with neonates to effect positive development rather than specifically about pain.
p 6 line 221 typo rhythmically
p 6 paragraph starting line 224. This paragraph feels as if it makes broad statements and I wonder how these backed up by evidence? For example, 'music expression fosters contection of a relationship that may have important implications for children’s endurance of other kinds of pain in later hospitalizations, particularly when experiencing an acute pain crisis' - really? how does music expression foster that connection? Compared to previous writing which feels very factual, I wonder if a medical reader might feel this has gone too far into subjective assumptions about unconscious processes. I would suggest trying to be more concrete here. There are several small typos in this paragraph also.
p6 line 234 Whereas (should be small w?)
p10 line 337 - 340 There appears to be text missing here so the sentence doesn't make sense.
line 380 - 386 - I think this needs to go in section on neonatal pain not in the conclusion.
Nice concluding sentence. Is there a recommendation you might add regarding future research that is needed in the music and pain field?
Author Response
Thank you for this nice review.
I have answered point-by-point below:
2.2.1 Gate contro (typo need l)
+I added the l
p4 line 119 A pain experience is often linked to anxiety - reference this claim
+I added a large recent study-by de Heer et al Good idea-thanks!
p4 line 132 - 133 I wonder about just using the terms active and receptive rather than passive? Do you need both terms passive and receptive?
++since many have brought these terms into the fold and they imply different things-I;'d like to leave them
p 5 paragraph starting line 164. The use of 1st person here is jarring. Up until now the whole section is factual and descriptive. Suddenly the author says 'I have...' I wonder if this should come later, or be written in 3rd person still to match this whole section which is presenting evidence and prior information.
+++AMA and a growing number of journals have recommended when discussing one's own work to implement the "I"-so I think I will leave it
2.4.1 neonatal pain
While I can see how it makes sense to start with neonates and work through the age groups, this section feels a little out of place. There is rare mention of pain in this section, it seems more about working with neonates to effect positive development rather than specifically about pain.
+++Since neonates cannot use descriptions to speak of their pain-this topic does not get much attention...it is important that they be represented-because infant expression are all musical...as such, I have added a pain study-the work of Ullsten and her colleagues. Good idea. Thank you.
p 6 line 221 typo rhythmically
+++fixed. Thank you.
p 6 paragraph starting line 224. This paragraph feels as if it makes broad statements and I wonder how these backed up by evidence? For example, 'music expression fosters connection of a relationship that may have important implications for children’s endurance of other kinds of pain in later hospitalizations, particularly when experiencing an acute pain crisis' - really? how does music expression foster that connection? Compared to previous writing which feels very factual, I wonder if a medical reader might feel this has gone too far into subjective assumptions about unconscious processes. I would suggest trying to be more concrete here. There are several small typos in this paragraph also.
+++I fixed the typos. In answer to your question-yes, really. I do mention Erikson, and provide a developmental basis of trust. However, rather than simply theory-and experience, which is a second and third reference-I added herein another study of healthcare providers and this provides the evidence and context for the above claim-about children, pain and trust.(DeCcosta study and added quote from that work)
p6 line 234 Whereas (should be small w?)
+I fixed this typo and made it into a sentence by adding a period
p10 line 337 - 340 There appears to be text missing here so the sentence doesn't make sense.
++I fixed the tenses so that the sentence now makes sense. Thank you.
line 380 - 386 - I think this needs to go in section on neonatal pain not in the conclusion
+++Actually, I leave it here because it is emphasizing what is left out of the study of pain in general. It is an isszue that applies to adults (carers), neurologists, and neonatologists and it sheds light on the fact that we are still way behind caring for all humans-and believing their pain.
Nice concluding sentence. Is there a recommendation you might add regarding future research that is needed in the music and pain field?
+++I have tightened up the conclusion to include the musical aspects and future potential. Thank you.
Reviewer 2 Report
Excellent manuscript. Thank you for writing such an important and valuable paper. The following are my questions/comments/suggestions:
1. Please review the paper as there are several punctuation issues, type-o's (i.e. Gate Contro), inconsistent verbs in a list (i.e. lines 158-161), and missing possessives (i.e. line 193 should be infants').
2. Lines 40-41 - "the patient's team of medical and involved family members" - should it say "medical professionals"? As it reads now it seems that the family members also have a medical background.
3. 2.1. Definition of Pain - first paragraph - do you need a citation here?
4. 2.3 Types of Pain - first paragraph - do you need citations where the definitions are provided, or is this considered to be common knowledge?
5. Lines 136-137 - different music therapists may have a different definition of "passive music therapy". You mentioned that it is "without the involvement of a music therapist". To me, if that is the case, is it still music therapy? When I've worked with patients as a music therapist, I've considered it passive music therapy if the patient is just listening to me play live (or recorded) music for them and they are minimally engaging with me. Just something to consider.
6. Lines 166-167 - "tonal intervallic synthesis" - could you provide more of an explanation of what this is?
7. Lines 194-196 - please provide the citation for this sentence so we know to which researchers you are referring.
8. Line 203 - what do you mean by "song of kin"?
9. Lines 234-236 - I believe this should be one sentence starting with "Whereas" and ending at the end of the paragraph. Otherwise, it does not make as much sense.
10. Lines 246-248 - please review the sentence that begins with "Concerns related..." as it does not read well.
11. Line 279 - do you truly mean "sex" or do you mean "gender"?
12. 2.4.5 Palliative care - this sections seems more about sleep than it does palliative care. The final paragraph focuses primarily on palliative care, and it does not seem to fit with the rest of the section.
13. Please double check your references. For example #37 & #38 seem to be missing the title of the articles. Also should #50 list the American Academy of Pediatrics as the author at the beginning of the reference?
Author Response
Thanks you for your positive critique and your careful attention to my article
- Please review the paper as there are several punctuation issues, type-o's (i.e. Gate Contro), inconsistent verbs in a list (i.e. lines 158-161), and missing possessives (i.e. line 193 should be infants').
+++Thanks you-I have added the l in "control" and fixed the grammar. The "infant's" is correct.
2. Lines 40-41 - "the patient's team of medical and involved family members" - should it say "medical professionals"? As it reads now it seems that the family members also have a medical background.
+++Good point. Thank you. I fixed this to include medical and psychosocial professionals.
3. 2.1. Definition of Pain - first paragraph - do you need a citation here?
+++As the opening is speaking generally, I don't think straight away, that a definition/reference is needed. By the third sentence, I do include an important reference.
4. 2.3 Types of Pain - first paragraph - do you need citations where the definitions are provided, or is this considered to be common knowledge?
+++This is considered to be common knowledge
5. Lines 136-137 - different music therapists may have a different definition of "passive music therapy". You mentioned that it is "without the involvement of a music therapist". To me, if that is the case, is it still music therapy? When I've worked with patients as a music therapist, I've considered it passive music therapy if the patient is just listening to me play live (or recorded) music for them and they are minimally engaging with me. Just something to consider.
+++The problem is, that "passive" implies that the patient is not doing anything, as a result of listening. However, listening can be quite activating, so I would not necessarily define when a MT plays for, or uses a recording "passive". Also, another problem particularly in medical research, is that other medical professionals call "passive music therapy" -"music therapy" and it is not because they are not trained or certified music therapists.
6. Lines 166-167 - "tonal intervallic synthesis" - could you provide more of an explanation of what this is?
+++I have provided a reference and upon your suggestion have added a bit more description.
7. Lines 194-196 - please provide the citation for this sentence so we know to which researchers you are referring.
+++I made clear it was van Dokkum et al -Thank you!
8. Line 203 - what do you mean by "song of kin"?
+++I added a reference with a description preceding on 'song of kin'
9. Lines 234-236 - I believe this should be one sentence starting with "Whereas" and ending at the end of the paragraph. Otherwise, it does not make as much sense.
+++True! Somehow the coma, turned into a period, and so I have changed it back to reflect proper grammar usage. Thanks.
10. Lines 246-248 - please review the sentence that begins with "Concerns related..." as it does not read well.
+++I fixed the grammar of this sentence and it now flows more easily.
11. Line 279 - do you truly mean "sex" or do you mean "gender"?
+++sex
12. 2.4.5 Palliative care - this sections seems more about sleep than it does palliative care. The final paragraph focuses primarily on palliative care, and it does not seem to fit with the rest of the section.
+++Good point-I have moved the final paragraph on palliative care and pain, to the forefront, and provided a reference indicating the important, and often overlooked relationship between sleep and palliative care.
13. Please double check your references. For example #37 & #38 seem to be missing the title of the articles. Also should #50 list the American Academy of Pediatrics as the author at the beginning of the reference?
+++I fixed these references and have verified that the AAP reference is correct
Reviewer 3 Report
This manuscript covers the fascinating topic of mechanisms underlying the use of music for pain management. The emphasis on neurologic implications is welcome, as there has been a plethora of research on the neurologic influences of music and music-based interventions in the recent literature.
The author does a fine job of introducing the reader to music therapy and music medicine and to pain, in general. The approach is from a clinical viewpoint, rather than that of a researcher. The use of first person narrative in places affirms this assertion, but are not appropriate for this journal. The assessment process that is described is interesting, from a clinical perspective. However, this reviewer believes that this type of content would be best suited in a textbook chapter, as there is no original research included, nor is there a thorough review of current evidence presented here.
The abstract and introduction begin with a discussion of chronic pain, but the article actually speaks as much to neonatal, pediatric procedural pain, and palliative care, as adult pain resulting from various diagnoses. The overview is quite broad, and misses the opportunity to present the reader with a wealth of scientific writing on the subject. There are so many articles on music for pain that researchers have published several meta-analyses and systematic reviews of the literature, relative to specific types and conditions of pain. For example, some of these important references include:
Music-Induced Analgesia in Chronic Pain Conditions: A Systematic Review and Meta-Analysis.
Pain Physician. 2017 Nov;20(7):597-610.
Lin, C., Hwang, S., Jiang, P., & Hsjung, N. (2020). Effect of music therapy on pain after orthopedic surgery: A systematic review and meta-analysis. Pain Practice, 20(4), 422-436. doi: http://dx.doi.org.ezproxy.uttyler.edu:2048/10.1111/papr.12864
Sibanda, A., Carnes, D., Visentin, D., & Cleary, M. (2019). A systematic review of the use of music interventions to improve outcomes for patients undergoing hip or knee surgery Journal of Advanced Nursing, 75, 502-516.
Lee, J. (2016) The effects of music on pain: A meta-analysis. Journal of Music Therapy, 53, 430-477.
Kuhlmann, A., Rooji, A., Lorses, :, can Kijk, M., Hunink, M. & Jeekel, J. (2018). Meta-analysis evaluating music interventions for anxiety and pain in surgery. BJS, 105, 773-783.
Hole, J., Hirsch, M., Ball, E., Meads, C. (2015). Music as an aid for postoperative recovery: A systematic review and meta-analysis, The Lancet, 386, 1659-1671.
In addition to these, there are several thoughtful papers on the topic that should be included, for example:
Lunde SJ, Vuust P, Garza-Villarreal EA, Vase L. Music-induced analgesia: how does music relieve pain? PAIN 2019;160:989–93. [PubMed] [Google Scholar]
Music-Induced Analgesia in Chronic Pain Conditions: A Systematic Review and Meta-Analysis.
Garza-Villarreal EA, Pando V, Vuust P, Parsons C. Pain Physician. 2017 Nov;20(7):597-610.
Music's objective classification improves quality of music-induced analgesia studies.
Martin-Saavedra JS, Saade-Lemus S. Pain. 2019 Jun;160(6):1482-1483. doi: 10.1097/j.pain.0000000000001535.
Does music ease pain and anxiety in the critically ill?
Chlan L, Halm MA. Am J Crit Care. 2013 Nov;22(6):528-32. doi: 10.4037/ajcc2013998. PMID: 24186825
The account of entrainment is most relevant as an innovative, potential music-based intervention for pain. However, the author fails to reference other articles on the topic of entrainment, including:
Kim, S, Gäbel,C, Aguilar-Raab,C, Hillecke, T, Warth, M, Affective and autonomic response to dynamic rhythmic entrainment: 430 Mechanisms of a specific music therapy factor, The Arts Psychother, Vol 60, 2018, 48-54, ISSN 0197-4556, 431 https://doi.org/10.1016/j.aip.2018.06.002.
Vuilleumier, P. & Trost, W. (2015). Music and emotions: from enchantment to entrainment. Ann. NY Acad. Sci. 1337, 212-222.ISSN 0077-8932
The author does reference “Beaty, R. E., et al., Cerebral Cortex, Vol. 31, No. 10, 2021” but as the details are incomplete (no title provided), it is not known whether this research is relevant to the argument.
In conclusion, this submission is an interesting introduction/overview of music interventions for pain. However, it does not live up to its title, as it fails to provide a scholarly presentation of underlying mechanisms or relevant current research.
Author Response
Reviewer 3
Thank you for pointing out that the topic is of interest and that the article’s introduction is timely.
- “The use of first person narrative in places affirms this assertion, but are not appropriate for this”
Response:
This article is not a clinical trial. It is defining how pain has been understood in theory and practice historically and how its focus in research, is applicable to clinical practice. According to the AMA Manual of Style:
“In general, authors should use the active voice, except in instances in which the actor is unknown or the interest focuses on what is acted on.” Most biomedical journals require the active voice.
http://www.biomedicaleditor.com/active-voice.html#:~:text=Emphasize%20the%20Active%20Voice&text=For%20example%2C%20the%20American%20Medical,on%20what%20is%20acted%20upon.%22
- “The assessment process that is described is interesting, from a clinical perspective. However, this reviewer believes that this type of content would be best suited in a textbook chapter, as there is no original research included, nor is there a thorough review of current evidence presented here.”
Response:
Thank you for mentioning the assessment process as I offer it, as a model is “interesting”-assessment is a critical miss in evaluating pain accurately in healthcare. The reviewer’s observation that the article’s tone is one of a clinician, rather than one of a researcher is not accurate. The perspective I present is both clinical and research-based. I believe the article does due diligence in providing a a blending of clinical-researcher perspective starting with why, particularly with pain, it is important to understand the history of how pain’s etiology has been addressed from multiple perspectives. This illustrates the concept of integration, which is emphasized in the title and brought in as a theme throughout the text. I address how the definition of pain has changed. Assessment is key and one of the most troublesome aspects of medical treatment is the under-diagnosis or mis-diagnosis of pain. This is why a large part of this introduction focuses on how critical it is to assess pain from several vantage points. This fits with the scope of this journal -which in print is inclusive of theoretical results, as described herein from the website: “Our aim is to encourage scientists to publish their experimental and theoretical results in as much detail as is required to fully convey the information”- I do find it appropriate that ASSESSMENT is inclusive of the quadrant and it preempts the model I propose which, later includes treatment options, with references from clinical practice.
- “The abstract and introduction begin with a discussion of chronic pain, but the article actually speaks as much to neonatal, pediatric procedural pain, and palliative care, as adult pain resulting from various diagnoses.”
Response:
I agree-I should not have confined the opening of the abstract or introduction confining the article’s topic to “chronic pain” I have changed this. This is an important change, and I am grateful for the reviewer’s catch of this. The term chronic, as opposed to acute, neuropathic, or nociceptive implies a pain experience based on duration/etiology. Pediatric patients and palliative patients can have enduring pain which is chronic-so those catgory/subheadings do not conflict with types of pain, as a category. I did define varying kinds of pain in the ‘Types of Pain’ section.
- You have suggested that the article should be more comprehensive of former pain research and that I include more studies and references.
Response:
- So few of the studies mentioned in the articles you’ve suggested -largely reviews- incorporate live music. Nevertheless, I took time to go through your recommendations.
This one-you’ve mentioned twice:
Garza-Villarreal EA, Pando V, Vuust P, Parsons C. Pain Physician. 2017 Nov;20(7):597-610.
I have added it and its findings-which include the fact that: the patients included mostly listened to music, with only one study using active singing, and 2 studies using live music.
1. The Hole et al article:
Hole, J., Hirsch, M., Ball, E., Meads, C. . Music as an aid for postoperative recovery: A systematic review and meta-analysis, The Lancet, 2015, 386, 1659-1671.
This is the largest review of music in post surgical pain and it does provide some music mechanisms which are interesting, though not surprising. Nevertheless, I have added their points into the article-with this paragraph, that is acknowledging the music mechanisms of their summary. Below is the added text.
“The largest review of music’s effect on procedural pain recovery for adults was undertaken in 2015, and in its evaluation of mechanisms, it acknowledged that pain experience is affected by both physical and psychological influences. The review indicates that listening to music can alter the actual intensity and unpleasantness of pain-which in turn can reduce the sensation of pain to be reduced. Additional potential mechanisms could result in reduced autonomic nervous system activity, such as reduced pulse, respiration rate, or decreased blood pressure.”
- Your review suggested I reference this article, and I already had it in there on page 5 paragraph 3, with mention of analysis:
Kim, S, Gäbel,C, Aguilar-Raab,C, Hillecke, T, Warth, M, Affective and autonomic response to dynamic rhythmic entrainment: 430 Mechanisms of a specific music therapy factor, The Arts Psychothery, Vol 60, 2018, 48-54, ISSN 0197-4556, 431 https://doi.org/10.1016/j.aip.2018.06.002.
- The reviewer suggested addition of Lunde SJ, Vuust P, Garza-Villarreal EA, Vase L. Music-induced analgesia: how does music relieve pain? PAIN 2019;160:989–93
While informative, this work is flawed. A commentary by the authors below, shed light on its flaws related to the over-generalization of ‘genre’ and even ‘composer’ in applications for treatment.
I added it-and defined what we learned from its limitations without incriminating the work you recommended.
Thank you for suggesting my purview of this piece, which led to a greater detailed discussion of musical decision-making in pain treatment.
Martin-Saavedra, J; Saade-Lemus, S. Music's objective classification improves quality of music-induced analgesia studies. PAIN: June 2019 – Vol 160, Issue 6 - p 1482-1483 doi: 10.1097/j.pain.0000000000001535
I also added findings of Gutsgell for this reason-because her team’s study discusses implementation of the distinct music therapy mechanisms related to centering and succession of distinct layering of sounds (ocean drum for breathing), choice, meter shifts, and key modes (mixolydian to start)-rather than prescriptive music named broadly and in an undefined way.
Added -with qualification of rationale for music selection.
Gutgsell, K. J., Schluchter, M., Margevicius, S., DeGolia, P. A., McLaughlin, B., Harris, M., … Wiencek, C. (2013). Music therapy reduces pain in palliative care patients: A randomized controlled trial. Journal of Pain & Symptom Management, 45(5), 822–831
Round 2
Reviewer 3 Report
The author's response to concerns is noted. This revision now includes several references and other editorial changes that were recommended. While these improve some of the arguments articulated in the paper, there is little additional support for the content promised in the title. Beyond the neuromatrix theory of pain, there is insufficient original thinking or additional research presented in this article to support a novel approach to "underlying music mechanisms influencing the neurology of pain." More technical, thorough, and contemporary analyses are expected in a contribution to "Brain Sciences."
